# Formulation and Characterization of Sodium Caseinate/Phloretin Complexes as Antioxidant Stabilizers in Oil-in-Water Emulsions

**DOI:** 10.3390/foods14020236

**Published:** 2025-01-14

**Authors:** Najme Kheynoor, Jean-Christophe Jacquier, Mohammadreza Khalesi, Amir Mohammad Mortazavian, Mohammad-Taghi Golmakani

**Affiliations:** 1Department of Food Science and Technology, School of Agriculture, Shiraz University, Shiraz, Iran; najme.kheynoor@shirazu.ac.ir; 2UCD Institute of Food and Health, School of Agriculture and Food Science, University College Dublin, Dublin, Ireland; mohammadreza.khalesi@ul.ie; 3Department of Biological Sciences, Faculty of Science and Engineering, University of Limerick, Limerick, Ireland; 4Department of Food Science and Technology, National Nutrition and Food Technology Research Institute, Faculty of Nutrition Sciences and Food Technology, Shahid Beheshti University of Medical Sciences, Tehran, Iran; mortazvn@sbmu.ac.ir

**Keywords:** antioxidant emulsifiers, oxidative stability, phloretin, protein-polyphenol interaction, sodium caseinate

## Abstract

Emulsifiers with antioxidant properties, such as protein/polyphenol complexes, adsorb at the oil-water interface and improve the physical and oxidative stability of emulsions. Here, 2% (*w*/*w*) sodium caseinate and varying concentrations of phloretin (0–10 mM) were used to stabilize oil-in-water emulsions. Control emulsions with protein alone showed poor stability with increased droplet sizes from 0.33 µm to 5.18 µm after 30 days, while no significant change was observed in emulsions containing phloretin (remaining below 400 nm). The in vitro antioxidant activities increased with increasing phloretin concentrations (0 to 10 mM). In the ABTS assay, the antioxidant activity improved from 14.02 ± 8.33% to 95.09 ± 1.31%, and in the DPPH assay, it increased from 32.59 ± 2.73% to 99.03 ± 0.14%. Similarly, the oxidative stability of the emulsions improved with increasing phloretin concentrations (0 to 10 mM). After 30 days of storage, PV decreased from 38.22 ± 2.58 µM to 11.81 ± 2.55 µM, and MDA content reduced from 48.43 ± 0.31 µM to 7.24 ± 0.21 µM. Measuring the apparent viscosity demonstrated a reduction in viscosity with the addition of phloretin. These findings demonstrate that incorporating phloretin into sodium caseinate-stabilized emulsions as a novel antioxidant emulsifier can be an effective strategy to extend the shelf life of emulsified food products prone to oxidative deterioration.

## 1. Introduction

Food emulsions consist of at least two incompatible phases and, based on their dispersion phase, are divided into oil-in-water (O/W) and water-in-oil (W/O) emulsions [1]. O/W emulsions play a crucial role in many popular commercial food products (e.g., cream and milk). Emulsifiers, surfactants, and some macromolecules are used for the stabilization of emulsions by reducing surface tension and coating the surface of the droplets [2]. Proteins as surface-active molecules can be used as emulsifiers to produce O/W emulsions with desirable physicochemical properties [3]. Proteins are amphiphilic polymers that can adsorb onto oil/water interfaces and form an interfacial film around the droplets and, therefore, can be used as emulsifiers to produce O/W emulsions with desirable physicochemical properties [3]. There are various types of functional groups on the protein surface that can bind to different types of small molecules through hydrogen bonding, hydrophobic interactions, and electrostatic interactions [4,5]. As such, Protein/polyphenol complexes have gained attention for their excellent properties in stabilizing emulsions as novel antioxidant emulsifiers while also encapsulating nutrients with health benefits. Incorporating polyphenols into proteins as emulsifiers can enhance the physical and oxidative stability of emulsions and improve their resistance to in vitro digestion [6].

Sodium caseinate, a non-globular protein, is widely used in the food industry due to its availability and low cost. It is commonly incorporated into food formulations as an additive, providing biological effects and impressive emulsifying properties [7]. However, like many protein-based emulsifiers, sodium caseinate alone may not provide sufficient protection against oxidation. In recent years, sodium caseinate/polyphenol complexes have been investigated as stabilizers, including sodium caseinate/epigallocatechin-3-gallate (EGCG) [8], sodium caseinate/tannic acid [7], sodium caseinate/EGCG/resveratrol [9]. Also, the effect of polyphenol structures on the antioxidant activity of sodium caseinate combined with various polyphenols (apigenin, kaempferol, phloretin, (+)-catechin, coumarin, *p*-hydroxybenzoic acid, resveratrol, hydroxytyrosol, and *p*-coumaric acid) investigated and reported sodium caseinate complexes with catechin, resveratrol, phloretin, and hydroxytyrosol can act as novel *antioxidant emulsifiers* to stabilize the emulsion [10].

Phloretin or 3-(4-Hydroxyphenyl)-1-(2,4,6-tri hydroxyphenyl) propane-1-one is a natural antioxidant found in apple leaves, Manchurian apricots and the bark of some fruit trees [11,12]. This natural phytochemical is a dihydrochalcone with the molecular formula C15H14O5 and a molecular weight of 274.3 g/mol, with low solubility in water and high solubility in organic solvents. Phloretin exhibits several health-promoting properties, including antidiabetic, antioxidant, anti-inflammatory, and antitumor effects [11], making its incorporation into foods an effective strategy to enhance or maintain human health. Nevertheless, its widespread use in the food, cosmetic, and pharmaceutical industries is restricted by inadequate water solubility, chemical stability, and low bioavailability. One effective approach to address these challenges is using colloidal delivery systems, such as micelles, microemulsions, nanoemulsions, emulsions, solid lipid nanoparticles, nanostructured lipid carriers, biopolymer nanoparticles, and microgels [13]. Various colloidal delivery systems can be developed using food-grade biopolymers, including proteins and polysaccharides. He et al. [14] reported that the bio-accessibility of encapsulated phloretin increased with gliadin/sodium carboxymethyl cellulose nanoparticles. Gong et al. [10] introduced sodium caseinate/phloretin complexes as a novel antioxidant emulsifier, highlighting that phloretin can enhance the antioxidant activity of sodium caseinate particles. Encapsulation systems utilizing food-grade proteins have shown great potential for preserving the functional properties of phloretin, owing to their natural origin and high nutritional value, which are well-received by consumers. However, there is limited research on the development of colloidal complexes using food-grade proteins for the encapsulation of the bioactive compound phloretin [13,14]. However, the use of phloretin for stabilizing emulsions has been barely reported, presenting a potential area for further exploration.

This study aimed to produce emulsions using sodium caseinate/phloretin complexes. The effects of phloretin as an antioxidant on sodium caseinate were investigated at various concentrations. Additionally, the interaction between sodium caseinate/phloretin complexes and their particle size was measured. Furthermore, the particle size, physical stability, oxidative stability, emulsifying properties, and rheological properties of the emulsions were analyzed. This work proposes a potential strategy to enhance the antioxidant activity of sodium caseinate, thereby improving the oxidative stability of emulsions and creating multifunctional emulsifiers with health benefits.

## 2. Materials and Methods

### 2.1. Materials

Sodium caseinate (Armor Proteins, Loudeac, France), phloretin (98%, green apple extract powder, Herba Diet Co., Rohtak, India), and sunflower oil (local supermarket, Dublin, Ireland) were used in this study. All additional reagents were analytical grade.

### 2.2. Preparation of Sodium Caseinate/Phloretin Complexes

Sodium caseinate/phloretin complexes were prepared according to Gong et al. [10] with some modifications. Briefly, sodium caseinate powder (2%, *w*/*w*) was weighed and dissolved in deionized water, stirred magnetically at 4 °C overnight. Phloretin dissolved in acetone was added to the sodium caseinate solution at different final concentrations of 0, 0.5, 2, 4, 8, and 10 mM and stirred at room temperature for 3 h. Acetone was slowly removed under a vacuum hood with controlled volume while the solution was gently stirred to minimize concentration changes. After removing acetone, the samples were stored at 4 °C overnight to promote complete interaction between sodium caseinate and phloretin.

### 2.3. Characterization of Sodium Caseinate/Phloretin Complexes

#### 2.3.1. Particle Size Measurement

Particle size and polydispersity index (PDI) of solutions were characterized using a Zetasizer Nano ZSE (Malvern, UK). The refractive index of water was 1.333. Sodium caseinate refractive index and absorption were 1.590 and 0.010.

#### 2.3.2. UV/Vis Spectroscopy

Sodium caseinate/phloretin solutions were diluted ×1000 with deionized water, and the UV/Vis spectrometer wavelength was recorded within the wavelengths of 240 nm to 360 nm (Shimadzu, UV-mini 1240, Kyoto, Japan).

#### 2.3.3. In Vitro Antioxidant Activity

##### DPPH Radical Scavenging Activity

The DPPH scavenging activities of sodium caseinate/phloretin complexes were evaluated using a modified method described by Gong et al. [10]. DPPH stock solution was prepared in methanol and diluted to reach an initial absorbance of around 1.2 at 515 nm. Then, 1 mL DPPH stock solution was thoroughly mixed with 0.2 mL of the sample and allowed to react at room temperature in the dark for 30 min. The absorbance was measured at 515 nm, using methanol as a reference. The DPPH scavenging activity was determined using the following equation:DPPH scavenging activity (%) = (A_0_ − A_1_)/A_0_ × 100(1)
where A_1_ and A_0_ are the absorbance of the sample and the control, respectively.

##### ABTS Radical Scavenging Activity

ABTS radical scavenging activity measurement was determined based on the approach provided by Meng et al. [15]. Briefly, a volume ratio of 1 to 2 was used to combine potassium persulfate aqueous solution (2.45 mmol/L) with ABTS aqueous solution (7 mmol/L). After that, it was incubated in the dark at room temperature for 12 h. Adjusting the mixture with methanol to obtain absorbances of 0.7 ± 0.02 at 734 nm. Then, 4 mL of ABTS solution was mixed with 0.1 mL of sample solution and then vigorously vortexed. Subsequently, the absorbance of the sample solution was measured at 734 nm. The ABTS radical scavenging activity was determined using the following equation:ABTS scavenging activity (%) = (A_0_ − A_1_)/A_0_ × 100(2)
where A_1_ and A_0_ are the absorbance of the sample and the control, respectively.

### 2.4. Emulsion Preparation

Freshly fabricated sodium caseinate/phloretin complexes solution and oil were mixed with the 80/20 (*w*/*w*) ratio and magnetically stirred (700 rpm for 2 min). High-pressure homogenization (GEA Niro Soavi model Panda Plus 2000, Parma, Italy) was used with 6 and 60 MPa for the first and the second stages, respectively, with three passes.

### 2.5. Characterization of Emulsions

#### 2.5.1. Emulsifying Properties

The emulsifying properties of emulsions were assessed using the emulsifying activity index (EAI) and emulsion stability index (ESI). EAI was measured according to Guo et al. [16]. Briefly, 1 mL of emulsions was diluted ten times with deionized water, and then 1 mL of diluted emulsion was mixed with 9 mL of 0.1% SDS solution. Using SDS prevented the emulsion droplets from flocculation. Finally, the absorbance was measured at 500 nm at 0 (A_0_) and 10 min (A_10_) in a microplate reader (BioTek Instruments, Winooski, VT, USA). EAI and ESI were measured and calculated according to the equations:EAI (m^2^ g^−1^) = (2T × A_0_ × dilution factor)/(c × (1 − Φ) × L × 10,000)(3)ESI (%) = 100 − (EAI_0_ − EAI_10_/EAI_0_ × 100)(4)
where T is 2.303, the dilution factor is 100, c is the weight of protein per unit volume (g/mL) (0.02), L is the width of the optical path (0.01 m), and Φ is the oil volumetric fraction (0.2).

#### 2.5.2. Determination of Distribution Droplet Size

The distribution droplet size of emulsions was characterized using a Mastersizer 3000 (Malvern Instruments Co., Ltd., Worcestershire, UK). The refractive index of oil was 1.468, and the refractive index of water was 1.333. After production, measurements were made in triplicate.

#### 2.5.3. Droplet Size Stability

The centrifugal stability of emulsions was measured by the droplet size (D_4,3_ and D_3,2_) before and after centrifuging (3500 rpm for 20 min) using a Mastersizer 3000 in triplicate on days 0, 1, 3, 7, 10, and 30 at 4 °C according to Haney et al. [17] and Restu et al. [18].

#### 2.5.4. Physical Stability (Sedimentation and Creaming Index)

The sedimentation stability was eye-monitored after centrifuging and taking pictures. The emulsion creaming index (CI) was evaluated to measure the storage stability of emulsions against separation [19]. The samples were placed in the glass tubes and stored at 4 °C. The CI measured 1, 3, 7, 10, and 30 days after production.CI (%) = (H_S_/H_T_) × 100(5)
where H_S_ and H_T_ are the height of the serum layer (mm) and the total emulsion (mm), respectively.

#### 2.5.5. Oxidative Stability Measurement

The emulsions were collected in falcon tubes and incubated (BD 115, Binder, Germany) at 40 °C to measure their stability under accelerated storage conditions for 30 days.

##### Determination of Hydroperoxides

The PV of the emulsions was measured according to Ju et al. [20]. Briefly, the emulsions (0.3 mL) were mixed with 1.5 mL of a mixture of isooctane and isopropanol (3:1, *v*/*v*) shaken continuously for 10 s, then centrifuged at 2000× *g* for 5 min, and 200 µL of the upper layer was combined with a solution containing 2.8 mL of methanol/butanol mixture (2:1, *v*/*v*), 15 μL of NH_4_SCN (3.94 M), and 15 μL of divalent iron ion solution (0.132 M BaCl_2_ and 0.144 M FeSO_4_ mixed in a 1:1 ratio and passed through a 0.22 μm filter). After the reaction time (20 min), the absorbance of the mixture was evaluated at 510 nm using a UV spectrophotometer (UV-mini 1240, Shimadzu, Japan). PV in μmol/L of emulsions was determined using an external standard curve (hydrogen hydroperoxide).

##### Determination of Malondialdehyde

The measurement of malondialdehyde (MDA) was conducted following the method described by Ju et al. [20] and Zhu et al. [21] with some modifications. Thiobarbituric acid (TBA) test solution was prepared with 15% trichloroacetic acid and 0.375% TBA dissolved in 0.25 M HCl. Then, 2 mL of TBA solution was mixed with 1 mL of the samples. The mixture was heated in the water bath (15 min), cooled to room temperature, and centrifuged (2000× *g*, 15 min). The absorbance of the supernatant was determined at 532 nm, and the TBARS content was measured using a standard curve generated from 1,1,3,3-tetraethoxypropane (μmol/L of emulsions).

#### 2.5.6. Apparent Viscosity

On the day of production, the apparent viscosity of emulsions was studied at 25 °C and the shear rate range of 1 to 100 s^−1^ using a rheometer (Physica MCR 301, Anton Paar, Graz, Austria) equipped with the cup and bob. All samples were allowed to rest for 10 min for equilibration and structure recovery before the measurement. For each measurement, samples were carefully deposited in the rheometer cup and measured in triplicate. The flow curve (logarithmic (RheolabQC)) test was conducted to determine the apparent viscosity of emulsions.

### 2.6. Statistical Analysis

All measurements were conducted in triplicate. The analysis was achieved using SPSS 24 software. The mean ± standard deviation of data is presented. Variance (ANOVA) analysis was performed, and Duncan’s multiple range tests were used for mean comparison. Time-dependent data were analyzed using the repeated measurement test. The significant level was considered to be *p* < 0.05.

## 3. Results and Discussion

### 3.1. Characterization of Sodium Caseinate/Phloretin Complexes

#### 3.1.1. Particle Size Measurement

The effect of phloretin concentration on the formation of sodium caseinate/phloretin complexes was investigated. Figure 1 shows the size of sodium caseinate/phloretin complexes at different concentrations of phloretin. Adding phloretin slightly increases the size of caseinate particles, suggesting a concentration-dependent interaction between sodium caseinate and phloretin. The results showed that sodium caseinate particles alone were 225.53 ± 0.46 nm. By adding phloretin up to 10 mM, the particle size increases up to 262.47 ± 0.06 nm, which suggests enhanced interactions that could influence the functional properties of these complexes. The results of the PDI of complexes (Figure 1) show that increasing phloretin concentrations increases the PDI sightly, but in general, low PDI indicates that the complexes are monodisperse particles. Low PDI in sodium caseinate/phloretin complexes signifies a stable system that is advantageous for optimizing formulations in food products.

#### 3.1.2. UV/Vis Spectroscopy

The UV-Vis spectra of sodium caseinate/phloretin complexes are presented in Figure 2. The results show that the UV-Vis spectra of sodium caseinate are affected by different concentrations of phloretin. The optimum sodium caseinate and phloretin absorption were detected at a wavelength of 280 and 288 nm, respectively. The absorbance peak for sodium caseinate around 280 nm is related to the cumulative absorbance of aromatic amino acids (tryptophan, phenylalanine, and Tyrosine) present in the sodium caseinate [22]. By increasing phloretin concentration, the total peak area of the complex was increased significantly. A similar result was reported by Yanti et al. [23], who studied the interaction between phloretin and insulin and showed that increasing the phloretin ratio increased the peak absorbency of insulin. The maximum absorbance of the complex with 0, 0.5, 2, 4, 8, and 10 mM phloretin was 280, 280, 282, 284, 285, and 285 nm, respectively. The hydrogen bond between sodium caseinate and the carboxyl groups of phloretin causes red-shifted emissions. Similarly, Gong et al. [10] indicated that the binding force of sodium caseinate/polyphenol complexes might be intermolecular hydrogen bonds according to their FTIR spectra and molecular docking results, and they reported that two hydrogen bonds formed between sodium caseinate and phloretin. A significant red shift (from 280 to 285 nm) indicates that the polarity of sodium caseinate has been changed by bounding phloretin to it. Such a red shift is supported by the interaction study that the absorption of protein value at 280 nm increased when polyphenols were added, suggesting the changes in the protein structure after interaction with polyphenols [24].

#### 3.1.3. Antioxidant Activity

The DPPH and ABTS scavenging activities of sodium caseinate/phloretin complexes are shown in Figure 3. The DPPH radical scavenging activities of sodium caseinate/phloretin complexes containing 0 (control), 0.5, 2, 4, 8, and 10 mM samples were 32.59 ± 2.73%, 49.40 ± 7.21%, 63.73 ± 4.79%, 89.42 ± 0.64%, 98.89 ± 0.09%, and 99.03 ± 0.14%, and the ABTS antioxidant activities were 14.02 ± 8.33%, 51.65 ± 4.51%, 66.05 ± 5.39%, 73.59 ± 8.66%, 89.94 ± 2.66%, and 95.09 ± 1.31%, respectively. The results of both antioxidant activity tests indicated that when phloretin concentration increased from 0 to 8 mM, the antioxidant activity of sodium caseinate/phloretin nanoparticle solutions increased significantly. Still, the difference in antioxidant activity for 8 and 10 mM was not significant. Also, the antioxidant activity had a higher slope from concentration 0 to 4 mM, but from concentration 4 to 8 mM and from 8 to 10 mM, this slope became lower and lower. These results indicate a clear positive correlation between phloretin concentrations and the antioxidant activity of sodium caseinate/phloretin complexes, suggesting that higher phloretin content contributes to enhanced antioxidant capabilities. The significant increase in antioxidant activity can be attributed to the inherent properties of phloretin, which are well-known for its antioxidant potential. Polyphenol structure, especially the number and position of hydroxyl groups, affects the interaction of polyphenols with proteins and the antioxidant activity of their complexes [25]. As the concentration of phloretin increases, the availability of hydroxyl groups, which are crucial for radical scavenging, also increases. This enhancement in antioxidant activity may result from the formation of stable complexes between sodium caseinate and phloretin, which not only protect the polyphenols from degradation but also facilitate their interaction with free radicals. Thongzai et al. [26] used whey protein in combination with phenolic compounds such as gallic acid, ferulic acid, and tannic acid. They reported that DPPH and ABTS scavenging activities of complexes increased with an increase in the concentration of phenolics from 1% to 5. Another study showed that DPPH and ABTS scavenging activities were increased by combining chlorogenic acid with zein, β-lactoglobulin, and wheat gluten hydrolysate proteins [25]. When sodium caseinate was used with EGCG and resveratrol, the ABTS scavenging activity increased [9].

### 3.2. Characterization of Emulsions

#### 3.2.1. Emulsifying Properties

EAI and ESI of the sodium caseinate/phloretin complexes at different concentrations of phloretin are displayed in Figure 4. EAI, measuring the area of the oil/water interface covered per unit weight of protein, is related to the potential of proteins to cover oil droplets [27], which means how much sodium caseinate covers the interface area. ESI is related to the thickening of the interface layer of emulsions due to the presence of protein complexes [28]. The EAI of the sodium caseinate/phloretin-stabilized emulsions was decreased gradually by adding phloretin from 0.5 to 4 mM. Still, there were no significant differences between 4 and 8 mM of the phloretin and then increased significantly from 8 to 10 mM of the phloretin. The ESI of emulsions, contrary to EAI, increased by increasing the phloretin concentration to 4 mM of phloretin. Still, there were no significant differences between 4 mM and 8 mM of the phloretin and then decreased from 8 to 10 mM of the phloretin. The results show that increasing phloretin concentration could better cover the droplet of oils with sodium caseinate/phloretin complexes at the specific level of phloretin, so there is no regular relation between the covering of the oil droplets and the increase in phloretin concentration. As a result, in sodium caseinate/phloretin-stabilized emulsions at the specific content of phloretin, the coalescence and flocculation of oil droplets can be decreased effectively and make more stable emulsions. Tian et al. [29] reported similar results when increasing tea polyphenols in soybean protein isolate-stabilized emulsion; the EAI decreased but lacked regularity, and the ESI increased first until a particular concentration and then decreased by increasing tea polyphenols concentration. Li et al. [30] described that rice bran protein/catechin stabilized Pickering emulsion, there was not a regular relation between EAI and increasing catechin concentration, and ESI increased at the specific concentration and then decreased by additional catechin concentration. Also, Patil et al. [31] indicated that by increasing tannic acid to fish myofibrillar proteins, the EAI decreased, which can be related to the lower surface hydrophobicity of fish myofibrillar proteins, decreasing the diffusion and migration of particles to the interface due to the viscosity increase after tannic acid addition.

#### 3.2.2. Distribution of Droplet Size

The droplet size distribution of emulsions after production and after 30 days of storage is shown in Figure 5a,b, respectively. The emulsions with 0.5 and 8 mM phloretin had larger droplet size distribution after production, and the rest of the emulsions had narrow distributions close to each other. On the other hand, after 30 days of storage, the control emulsion had a shorter peak than other emulsions and also had two other peaks, indicating a broad and larger size distribution with significant polydispersity. All of the emulsions had narrow particle size distributions that showed high stability of sodium caseinate/phloretin-stabilized emulsions. This result can be attributed to the usage of a high-pressure homogenizer to make emulsions, which is related to the passage of emulsion droplets through a narrow gap at high speed that reduces droplet aggregation and improves emulsion stability [1,32].

#### 3.2.3. Droplet Size Stability Before and After Centrifuging

The surface-weighted mean (D_3,2_) and the volume-weighted mean (D_4,3_) of the emulsions before and after centrifuging are shown in Figure 5. D_3,2_ indicates the stability of emulsions at the interface, while D_4,3_ indicates the physical stability (flocculation and gravitational separation). D_3,2_ of emulsions before and after centrifuging are represented in Figure 5c,e, respectively. During the 10 days of storage, the largest droplet size of D_3,2_ related to emulsions containing 0.5 and 8 mM of phloretin with 60.2 and 55.43 nm (*p* < 0.05), respectively, but after 30 days of storage, the largest droplet size of D_3,2_ related to control emulsions with 64.3 nm (*p* < 0.05). D_4,3_ of emulsions before and after centrifuging are represented in Figure 5d,f, respectively. D_4,3_ of emulsions after 30 days of storage truly displayed the effect of phloretin on the physical stability of emulsions. D_4,3_ of control emulsions before and after centrifuging increased significantly to 5176.67 ± 130.13 nm and 987.67 ± 11.93 nm after 30 days of storage, while the emulsions containing phloretin did not change significantly at the same time except emulsion containing 10 mM of phloretin that increased significantly to 509.67 ± 16.26 nm after centrifuging. After centrifuging, the droplet size of emulsions decreased significantly, especially D_4,3_, which can be associated with reducing larger droplet size by centrifuging. Therefore, sodium caseinate/phloretin complexes improve the long-term stability of their emulsions by reducing droplet size. Some studies showed that complexes of protein/polyphenol as emulsifiers decreased the droplet size of emulsions: soybean protein isolate/tea polyphenols [29], rice bran proteins/catechin [30], soy protein-anthocyanin [20], sodium caseinate/inulin/konjac glucomannan [19], and sodium caseinate/gliadin nanoparticles [33]. They all reported adding or increasing polyphenols to protein increased the stability of emulsions due to the reduction of the droplet size of their emulsions.

#### 3.2.4. Physical Stability (Sedimentation and Creaming Index)

The sedimentation stability of emulsions after 30 days of storage and after centrifuging is shown in Figure 6b. Emulsions with 0 and 0.5 mM of phloretin had a few pale-yellow sediments that may refer to separating excess sodium caseinate from the system after centrifuging. This means that 2% sodium caseinate has been in excess of 0.5 mM of phloretin interaction. In emulsions with 2 and 4 mM of phloretin, there was no sedimentation, which indicates the proportionality of ratios of sodium caseinate and phloretin for interaction and stability. In emulsions with 8 and 10 mM of phloretin, there was some white sedimentation that can be related to the excess phloretin in the system without any strong bond. It shows that 8 and 10 mM of phloretin is high for interaction with 2% sodium caseinate for stabilized emulsions. Therefore, the ratio of protein and polyphenol is an essential key in emulsion sedimentation stability.

The CI indirectly evaluates the amount of droplet coalescence and flocculation in the emulsions since the higher the creaming rate, the larger the droplet size and the more unstable the emulsions [34]. All emulsions had no CI and were stable during 30 days of storage (Figure 6a). Even though after centrifuging (3500 rpm for 20 min), there was no separation phase and instability. Remarkably, the creaming behavior of emulsions was significantly affected by increasing particle concentrations at the interface of emulsions. The reason for this increase in the concentration of solid particles at the interface can be attributed to the emulsion production method (high-pressure). Similarly, Xu et al. [33] reported that decreasing the droplet size by high concentration of particles and increasing the viscosity of sodium caseinate/gliadin stabilized emulsions improved the emulsion creaming stability. Some studies reported an increase in the fraction of oil and particle concentrations, especially polysaccharides, by increasing the viscosity of emulsions, which are essential keys in the emulsion’s creaming stability [19,33]. In this study, we had very low viscous emulsions, but the size of droplets was shallow (<400 nm), and emulsions were stable during 30 days of storage. Therefore, the droplet size is the more effective key than viscosity and concentration in the stabilization of emulsions. A similar study reported that the high-pressure homogenization primarily affected the creaming behavior of O/W emulsions stabilized by sodium caseinate and showed that the most stable emulsions were gained by homogenization at high pressure, which can be attributed to the decreased emulsion droplet size [35].

Figure 6c represents the optical physical stability of emulsions at 4 °C after 60 days of storage. This picture exactly shows the effect of phloretin on the long-term stability of sodium caseinate/stabilized emulsions. The emulsion without phloretin was completely broken, and the oil was collected on the surface. Therefore, polyphenols have the potential to enhance the long-term physical stability of protein-stabilized emulsions.

#### 3.2.5. Oxidative Stability of Emulsions

Applying emulsifiers with antioxidant activity to fabricate emulsions is introduced as a strategy for improving oxidative stability. Combining protein and polyphenols as antioxidant solid particles can increase the physical and oxidative stabilities of emulsions [6]. Due to the excellent antioxidant activity of the sodium caseinate/phloretin complexes, they were used as antioxidants to stabilize emulsions. The oxidative stability of the emulsions was measured by monitoring PV value and MDA content during thermally accelerated for 30 days (Figure 7). The PV results (Figure 7a) and MDA results (Figure 7b) show that oxidation of emulsions without phloretin (control) was the highest after 30 days of storage, but by increasing phloretin concentration, the PV and MDA of emulsions reduced significantly. Therefore, increasing the phloretin concentration in the sodium caseinate/phloretin nanoparticle-stabilized emulsions promotes oxidative stability (*p* < 0.05). As the concentration of phloretin increased, the rate of PV and MDA formation in the emulsions slowed to the point where no significant increase in PV was seen for the first three days of storage when Phloretin was at or above 4 mM, and minimal change to the MDA level seen during 10 days of storage when Phloretin was at or above 8 mM. The rate of PV and MDA formation in the emulsions with 8 and 10 mM of phloretin were very close, which is a confirmation of the above antioxidant results. The slight increase of PV and MDA content in the presence of the phloretin can be related to the high-pressure homogenization, which increases the surface area by reducing the particle size, which increases the possibility of oxidation. Similarly, zein/tannic acid nanoparticles improved the stabilization of blackberry seed oils due to their antioxidant properties, and by increasing tannic acid concentration from 2 to 20 mg/mL, the PV and MDA content decreased significantly during 15 days of storage [36]. Zhao et al. [37] indicated that by the addition of gallic acid to zein nanoparticles, the oxidative stability of emulsion improved significantly.

#### 3.2.6. Apparent Viscosity of Emulsions

The viscosity of the emulsion represents its physical and chemical stability [30]. Figure 8 shows the apparent viscosity of sodium caseinate-stabilized emulsions without phloretin and with 10 mM of phloretin against shear rate. For both emulsions, the viscosity decreased slightly with the increase in the shear rate, which shows the shear-thinning behavior of emulsions. Similar results were reported in emulsion stabilized by chitosan and rice protein hydrolysate [38]. Shear thinning behavior is a characteristic of most emulsions that represents separating oil droplets from each other in the flocculated state [19,30]. Perrechil et al. [35] reported that sodium caseinate increased the electrostatic repulsive interactions between droplets in O/W emulsion. Therefore, phloretin may reduce the repulsive electrostatic interactions between droplets caused by sodium caseinate. Also, according to the intermolecular hydrogen bonds between sodium caseinate and phloretin [10], the hydrogen bond is easily reduced by the application of shear rate, so the viscosity decreases. Similarly, Hu et al. [39] reported that chitosan/pea protein stabilized emulsion had lower viscosity than pea protein-stabilized emulsion, which can be related to the hydrophilicity of the continuous phase. The amount of oil ratio phase used is an essential key in the food formulation. In many foods, including infant formula, the amount of oil phase should be less than 20%. By increasing the oil fraction followed by the increase in the viscosity of the dispersed phase, the stability of the emulsion increases, but with a low oil fraction, the stability of the emulsion is lower, which makes it more difficult to investigate its properties. Both emulsions in this study, with 20% oil, showed slight non-Newtonian behavior at low shear rates. Similarly, chitosan/rice protein hydrolysate stabilized O/W emulsion reported the same results for 20% oil fraction [38].

## 4. Conclusions

This study investigated the effect of different concentrations of phloretin on the physicochemical characteristics of sodium caseinate-stabilized emulsions. Sodium caseinate/phloretin-stabilized emulsions with high-pressure homogenization possessed high physicochemical stability. The UV/Vis study showed that increasing phloretin concentration increased the interaction. The sodium caseinate/phloretin-stabilized emulsions had an average diameter below 400 nm after 30 days without creaming instability. Sedimentation stability and ESI showed that 4 mM of phloretin is a better ratio than other concentrations to interact with 2% sodium caseinate. The emulsions acted as shear-thinning fluid, and the viscosity decreased under shear rate. The viscosity decreased by the addition of phloretin but the lower viscosity had no significant effect on the physical and oxidative stability of emulsions. The antioxidant activity of sodium caseinate/phloretin complexes was remarkably increased by increasing phloretin concentrations. Decreasing PV and MDA content of emulsions by increasing phloretin concentration during storage further reflected that the sodium caseinate/phloretin complexes have high antioxidant activity. This study can be proposed as a useful approach for the application of plant polyphenols for the bioactive delivery and long-term stabilization of emulsions in emulsified food products such as beverages.

## Figures and Tables

**Figure 1 foods-14-00236-f001:**
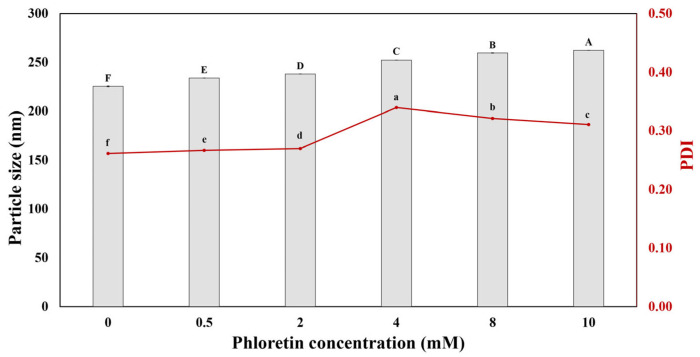
The particle size and polydispersity index (PDI) of sodium caseinate/phloretin complexes at different concentrations of phloretin. Different uppercase and lowercase letters indicate statistical differences between the samples for particle size and polydispersity, respectively (*p* < 0.05).

**Figure 2 foods-14-00236-f002:**
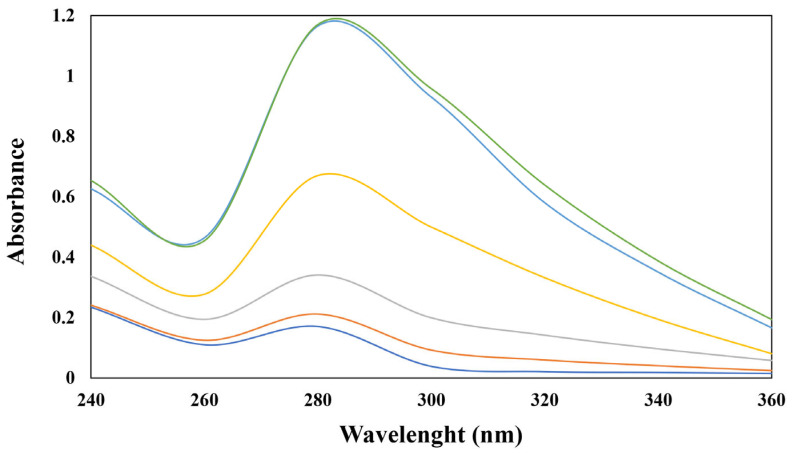
UV/Vis spectrum of sodium caseinate/phloretin complexes at different concentrations of phloretin. Control (
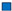
), 0.5 mM (
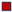
), 2 mM (
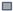
), 4 mM (
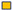
), 8 mM (
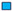
), 10 mM (
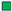
) phloretin.

**Figure 3 foods-14-00236-f003:**
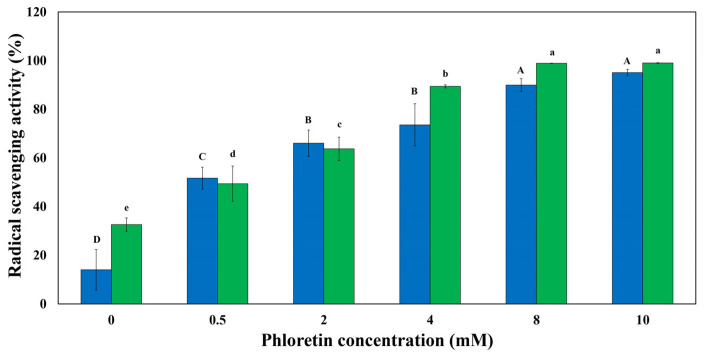
Antioxidant activity of sodium caseinate/phloretin complexes at different concentrations of phloretin: ABTS radical scavenging activity (blue color) and DPPH radical scavenging activity (green color); different uppercase and lowercase letters indicate statistical differences between the samples for ABTS and DPPH tests, respectively (*p* < 0.05). ([sodium caseinate] = 20 g/L).

**Figure 4 foods-14-00236-f004:**
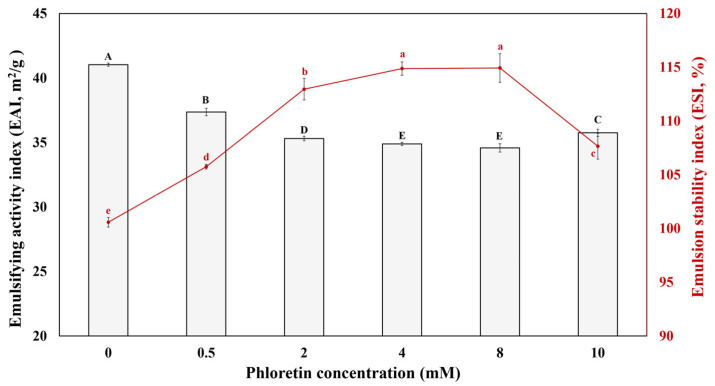
Emulsifying properties of sodium caseinate/phloretin complexes-stabilized emulsions at different concentrations of phloretin; different uppercase and lowercase letters indicate the difference between the samples for EAI and ESI tests, respectively (*p* < 0.05).

**Figure 5 foods-14-00236-f005:**
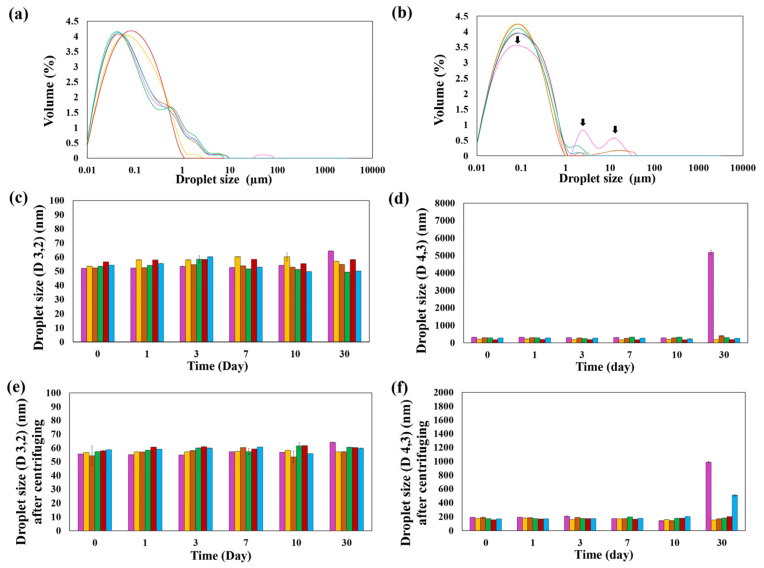
The droplet size distribution (**a**) after production, (**b**) after 30 days, and the mean droplet size of sodium caseinate/phloretin complexes-stabilized emulsions at different concentrations of phloretin during the storage before centrifuging: (**c**) D_3,2_ and (**d**) D_4,3_, and after centrifuging: (**e**) D_3,2_ and (**f**) D_4,3_ (*p* < 0.05). Control (
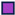
), 0.5 mM (
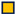
), 2 mM (
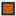
), 4 mM (
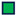
), 8 mM (
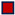
), 10 mM (
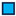
) phloretin.

**Figure 6 foods-14-00236-f006:**
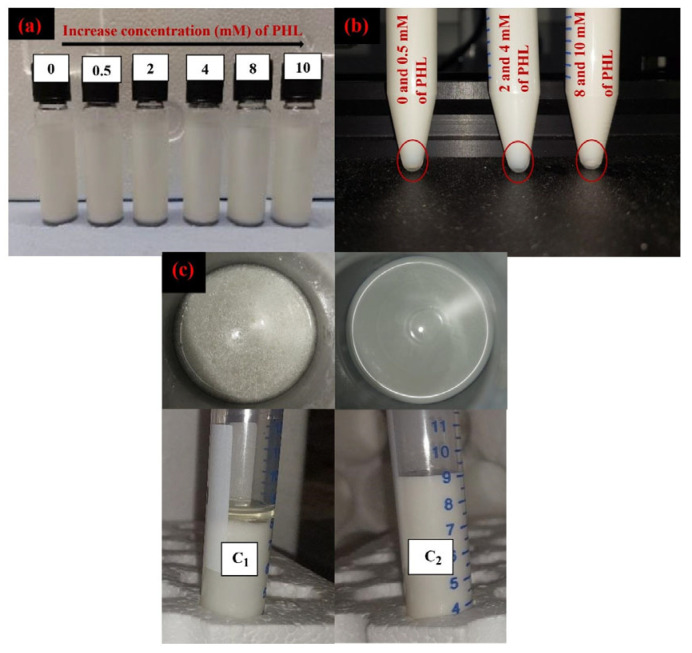
(**a**) Physical stability of sodium caseinate/phloretin complex stabilized emulsions after production, (**b**) Centrifuging sedimentation stability of emulsion samples after 30 days of storage, and (**c**) Physical stability of emulsions after 60 days of storage: (**c_1_**) without phloretin, (**c_2_**) with phloretin.

**Figure 7 foods-14-00236-f007:**
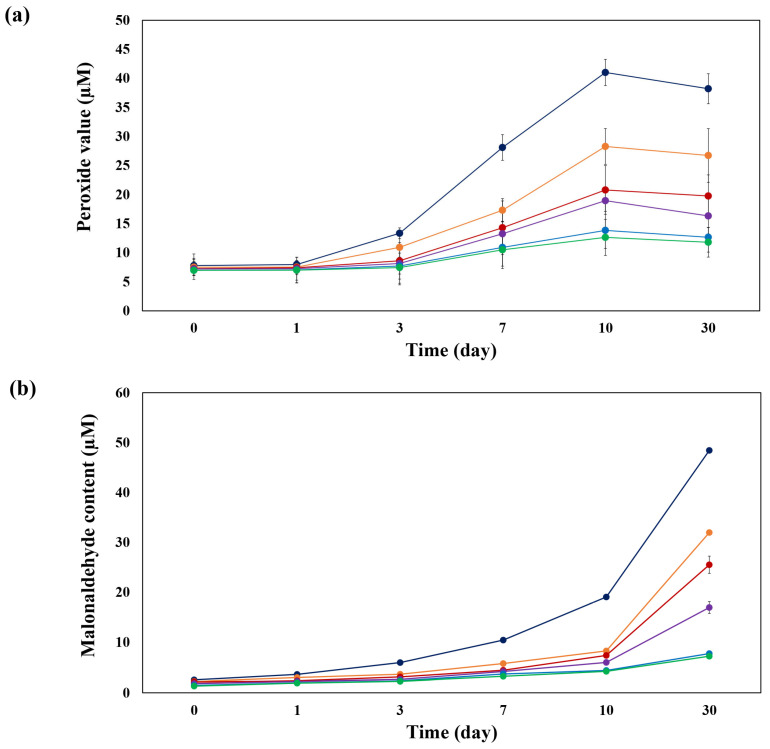
(**a**) Peroxide value and (**b**) malonaldehyde content of emulsions at different concentrations of phloretin during storage time (*p* < 0.05). Control (
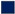
), 0.5 mM (
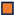
), 2 mM (
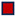
), 4 mM (
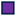
), 8 mM (
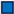
), 10 mM (
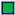
) phloretin.

**Figure 8 foods-14-00236-f008:**
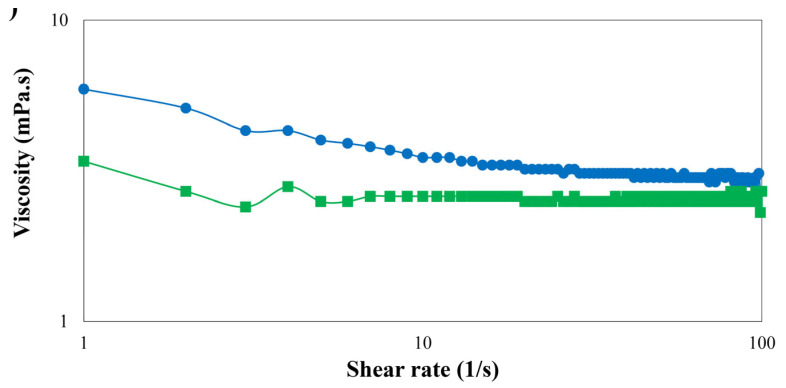
The viscosity versus shear rate for emulsions stabilized without phloretin (control (
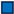
)) and 10 mM of phloretin (
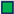
).

## Data Availability

The original contributions presented in the study are included in the article, further inquiries can be directed to the corresponding author.

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
