# Peer review of "Formulation and Characterization of Sodium Caseinate/Phloretin Complexes as Antioxidant Stabilizers in Oil-in-Water Emulsions"

_foods, 2025, doi:10.3390/foods14020236_

Round 1
Reviewer 1 Report
Comments and Suggestions for Authors
Dear Editor of food, thank you for the invitation to improve the quality of the manuscripts of your prestigious journal. I have reviewed the manuscript " Formulation and characterization of sodium caseinate/phloretin complexes as antioxidant stabilizers in oil-in-water emulsions" and consider it suitable for your journal after the authors make adjustments and respond point by point to each comment.
Comments to authors
1. In the introduction section, lines 40-41, include more information regarding why proteins function as emulsifiers. Also, give examples of proteins that have emulsifying properties, such as prolamins (zein, , sodium casinate, gelatin, etc. Cite (2019). Prolamins from cereal by-products: Classification, extraction, characterization and its applications in micro-and nanofabrication. Trends in Food Science & Technology, 90, 111-132.
2. In the introduction section, lines 59-61, include more information about the chemical structure of Phloretin, as well as physicochemical properties.
3. In Materials and Methods, section 2.1 Materials and 2.2. Preparation of sodium caseinate/phloretin complexes, the word sodium does not start with a capital letter. If it is by default, please capitalize the S in sodium.
4. In the results section, specifically in DPPH and ABTS antioxidant activity, mention the mechanisms (SET and HAT) by which phloretin and the complex give the antioxidant activity.
5. In the results section, specifically in "Apparent viscosity of emulsions" it mentions that the solutions behave as shear-thinning. To check this, I recommend performing the power law equation and showing in a table each component as K (Pa·sn), n and R2. (2024). Preparation and Characterization of Zein-Metformin/Gelatin Nanofibers by Coaxial Electrospinning. ACS omega, 9(37), 38423-38436.
Author Response
Thank you very much for taking the time to review this manuscript and kind comment. Please find the detailed responses below and the corresponding revisions/corrections highlighted in red in the re-submitted files

Reviewer 2 Report
Comments and Suggestions for Authors
The manuscript reports details on specific emulsifiers with antioxidant properties, namely protein/polyphenol complexes. The study is focused on sodium caseinate and varying concentrations of phloretin used to stabilize oil-in-water emulsions in view of applications to extend the shelf life of emulsified food products. The MS is very well-focused, properly detailed and contains important research results. This investigation is of considerable interest to the readers of MDPI Foods, and could be useful in view of further industrial applications.
However, before acceptance some minor issues should be addressed:
1. A list of the used abbreviations will be helpful, e.g. PV, TBARS, EGCG, CI, etc.
2. Some statements should be reformulated and/or clarified, e.g.
- lines 237-238 “…binding force of sodium caseinate/polyphenol complexes might be intermolecular hydrogen bonds.”;
- lines 283-284 “…the well potential of proteins to cover oil droplets…”;
-lines 407-408 “The rate of PV and MDA formation in the emulsions by increasing phloretin concentration became lower and lower significantly”
3. What does the statement “The emulsions acted as thinning fluid…” mean? (line 456)
Author Response

(The authors gave the same response as above.)

Reviewer 3 Report
Comments and Suggestions for Authors
Referee comments on the paper Formulation and characterization of sodium caseinate/phloretin complexes as antioxidant stabilizers in oil-in-water emulsions presented for publication in Foods.
The authors performed an analysis of emulsions containing a mixture of sodium caseinate/phloretin. The effects of phloretin as an antioxidant on sodium caseinate were investigated at various concentrations. The antioxidant properties of phloretin are very important from the point of view of producing food emulsions for health-conscious people. The authors indicate that the use of phloretin for the production of emulsions is not well documented in the literature and is a potential area for further research. I agree with this statement and treat it as an element of scientific novelty. The article is quite well constructed but I have a few doubts and comments before publishing it.
- Line 70, authors cite article 12 by providing the name of the first author, it should be cited He et al. [12]. This way of citing is used throughout the article. Multi-authored works should be cited according to the journal's instructions by providing the name of the first author and et al. Please check and correct in the entire text.
- Lines 90, 94, 109. We start sentences with a capital letter.
- Equation numbers 1 and 2 should be right aligned without the prefix Eq. Only the number in brackets (1) and (2).
- Why were the droplet distribution measurements conducted at 4°C and the rheological tests at 24°C?
- Were, for example, microscopic images of the emulsions taken to illustrate their structure?
- How long after preparing the emulsion were the rheological tests performed? From the point of view of emulsion stability this is quite important.
- Figures 3 and 4. I'm not entirely sure what the letters A, B, C and D in lowercase and uppercase mean on the graphs.
- Figure 5. In figures a and b I suggest changing the numerical notation on the y-axis from scientific to general (volume %). It will be more readable. And why are there negative values?
- The unit of dynamic viscosity is Pa·s, not Pa/s. In the shear rate unit, the second should be denoted with a lowercase "s." Additionally, the graph would look better if both axes were logarithmic. I also have some concerns regarding the rheological measurements. At low shear rates, fewer data points are provided compared to high shear rates, where there are significantly more. A more accurate characterization would be achieved if the frequency of measurement points were higher at low shear rates.
Why weren’t the curves for the remaining phloretin concentrations shown? For the 10% concentration, viscosity remains practically constant across the entire shear rate range, except for the first four measurement points. Classifying this as a non-Newtonian fluid is debatable, considering the low emulsion concentrations.
With these correction, the work will become suitable for publication in Foods.
Author Response

(The authors gave the same response as above.)
